# A macrophage-based screen identifies antibacterial compounds selective for intracellular *Salmonella* Typhimurium

Michael J. Ellis[1,2], Caressa N. Tsai[1,2], Jarrod W. Johnson [1,2,3], Shawn French [1,2], Wael Elhenawy[1,2], Steffen Porwollik[4], Helene Andrews-Polymenis[5], Michael McClelland[4], Jakob Magolan[1,2,3], Brian K. Coombes[1,2] & Eric D. Brown[1,2]

*Salmonella* Typhimurium (*S*. Tm) establishes systemic infection in susceptible hosts by evading the innate immune response and replicating within host phagocytes. Here, we sought to identify inhibitors of intracellular *S*. Tm replication by conducting parallel chemical screens against *S*. Tm growing in macrophage-mimicking media and within macrophages. We identify several compounds that inhibit *Salmonella* growth in the intracellular environment and in acidic, ion-limited media. We report on the antimicrobial activity of the psychoactive drug metergoline, which is specific against intracellular *S*. Tm. Screening an *S*. Tm deletion library in the presence of metergoline reveals hypersensitization of outer membrane mutants to metergoline activity. Metergoline disrupts the proton motive force at the bacterial cytoplasmic membrane and extends animal survival during a systemic *S*. Tm infection. This work highlights the predictive nature of intracellular screens for in vivo efficacy, and identifies metergoline as a novel antimicrobial active against *Salmonella*.

[1] Department of Biochemistry and Biomedical Sciences, McMaster University, 1280 Main St W, Hamilton, ON L8S 4K1, Canada. [2] Michael G. DeGroote Institute for Infectious Disease Research, McMaster University, 1280 Main St W, Hamilton, ON L8S 4K1, Canada. [3] Department of Chemistry and Chemical Biology, McMaster University, 1280 Main St W, Hamilton, ON L8S 4K1, Canada. [4] Department of Microbiology and Molecular Genetics, University of California Irvine, Irvine, CA 92697-4025, USA. [5] Department of Microbial Pathogenesis and Immunology, Texas A&M University, 8447 Riverside Pkwy, Bryan, TX 77807, USA. These authors contributed equally: Michael J. Ellis, Caressa N. Tsai. Correspondence and requests for materials should be addressed to B.K.C. (email: coombes@mcmaster.ca) or to E.D.B. (email: ebrown@mcmaster.ca)

The stagnant antibiotic discovery pipeline is particularly concerning for intracellular infections. Bacterial pathogens that survive within host cells evade the antimicrobial activity of the immune system and can form persister cells that tolerate antibiotic treatment[1,2]. As such, intracellular infections are often recurrent, warranting an increase in antimicrobial research specific for such pathogens and an improved understanding of the genetic requirements for intracellular survival. Intracellular bacteria occupy modified phagosomes (*Salmonella, Mycobacterium, Francisella*), inclusions (*Chlamydia*), lysosomes (*Legionella, Coxiella*), or the cytosol (*Listeria, Shigella, Ricksettia*) of host cells, which offer considerably different environments relative to standard nutrient-rich growth media[3]. Interestingly, emerging evidence suggests that pathogens such as *Staphylococcus aureus* and *Streptococcus pneumoniae* are also able to survive within host cells[4,5]. In these intracellular environments, genes that are otherwise dispensable for growth in nutrient-rich media often become essential, constituting a novel antimicrobial target space that is currently underexplored[6]. Genes conditionally essential within host cells may be overlooked in experimental systems that do not resemble the intracellular environment; indeed, recent systematic studies of the genetic requirements for growth in infection-relevant conditions revealed additional essential genes relative to those required for growth in vitro[7–9]. High-throughput screening platforms in conditions that closely resemble the intracellular environment have the potential to uncover novel antimicrobials that target conditionally essential genes.

*Salmonella enterica* serovar Typhimurium (*S*. Tm) is an intracellular pathogen and one of the leading causes of gastroenteritis worldwide[10]. Enteric salmonellosis is typically self-limiting in healthy individuals, although in the elderly and immunocompromised, *S*. Tm can invade into phagocytes for systemic spread through the reticuloendothelial system[11]. *Salmonella* infections are commonly treated with fluoroquinolones, cephalosporins, or macrolides[12], although cephalosporins do not penetrate phagocytic cells[13]. Unfortunately, resistance to these antibiotic classes is increasing worldwide[14,15]. Of further concern are extensively drug-resistant *S*. Typhi[16] and an invasive, multidrug-resistant variant of *S*. Tm that first emerged in sub-Saharan Africa (ST313)[17]. The emergence of antibiotic resistance across clades of *Salmonella* species threatens to intensify an already significant global health burden, underscoring the importance of novel antibiotic drug discovery.

During infection of macrophages and neutrophils, *S*. Tm occupies a modified phagosome called the *Salmonella*-containing vacuole (SCVs). A predominantly intracellular lifestyle affords protection from extracellular host immune defenses, including bile salts, antimicrobial peptides, and serum complement[11]. Host cells also act as reservoirs for dissemination to systemic sites, and often provide unique metabolic environments to shelter intracellular pathogens from nutrient competition with resident bacteria[18]. Several antimicrobial mechanisms intrinsic to the intracellular environment (i.e., metal depletion, vacuolar acidification, oxidative stress)[19] serve as environmental signals for multiple two-component regulatory systems in *S*. Tm that detect and respond to immune stresses with alterations in virulence gene expression[20].

In line with previous work suggesting discordance between in vitro and in vivo drug susceptibility[21], we hypothesized that *S*. Tm displays altered sensitivity to antimicrobials within macrophages. In particular, genes or processes that become essential within the intracellular environment may represent new drug targets or even sensitize *S*. Tm to existing or novel antibiotics. We therefore sought to screen an *S*. Tm ordered gene deletion collection[22] for growth impairment in media with ion availability

and pH resembling the intracellular vacuolar environment. These data and subsequent experiments with bacteria internalized in macrophages reveal that *S*. Tm is sensitized to the intracellular environment following genetic perturbation of metabolic and cell envelope biogenesis pathways. A high-throughput compound screen against intracellular *S*. Tm in macrophages identifies several small molecules with conditional efficacy only in acidic, ion-limited media, or macrophages, and not in nutrient-rich media. Interestingly, we also observe potentiation of canonically Gram-positive targeting antibiotics in the intracellular environment, which we ascribe to a loss of normal outer membrane structure in macrophage-internalized *S*. Tm. We report the identification of an intracellular-selective antimicrobial, metergoline, which disrupts the bacterial cytoplasmic membrane and prolongs animal survival in a murine model of systemic *S*. Tm infection.

## Results

**Screen for *S*. Tm genes required for intracellular growth.** In the systemic phase of infection, different populations of *S*. Tm encounter varying degrees of nutrient limitation and immune stressors, whether internalized within macrophages and neutrophils, persisting extracellularly in the bloodstream, or invading intestinal epithelial cells. We sought to identify growth-inhibitory small molecules that are specific for intramacrophage *S*. Tm, as macrophages are one of the primary host cell types manipulated by *Salmonella* for replication and systemic dissemination. We reasoned that to be selective for intracellular bacteria, a compound should interfere with one or more biological processes that are required only for growth in this environment, so we aimed to survey the genetic requirements for intracellular *S*. Tm growth. While others have identified *Salmonella* genes that become required (i.e., conditionally essential) for growth in conditions mimicking those in vivo[23–27]; to our knowledge, there has been no systematic, genome-scale survey of the impact of gene deletion on *S*. Tm survival in cultured macrophages.

The *S*. Tm str. 14028s genome contains ~4200 non-essential genes, ~3700 of which have been deleted in the ordered *Salmonella* single-gene deletion (SGD) collections[22,28]. Traditional macrophage infection assays are not practical for high-throughput screening with this large a number of individual strains, so we first aimed to identify SGD mutants with impaired growth in acidic, low-phosphate, low-magnesium media (LPM) that was established to resemble conditions in the SCV[29]. Importantly, gene expression in *S*. Tm grown in LPM and other types of acidic, ion-limited media has been shown to resemble that within macrophages[29–31]. We first measured the growth of 3725 SGD mutants in LPM, as a preliminary assessment of intracellular gene essentiality (Fig. 1a, Supplementary Data 1). This screen identified 125 genes important for growth in LPM, most of which are involved in nutrient biosynthesis and metabolism, as well as biological processes related to cell envelope homeostasis. Among others, this included amino acid (e.g., *aroE, hisA, argH, argG, serB*) and nucleotide (e.g., *pyrF, pyrE, purG, purE, purF*) biosynthesis, as well as LPS modifications and maintenance (e.g., *rfc, pgm, rfaK, rfaH, rbK, rfaI*).

We sought to verify the intracellular sensitivity of this LPM-sensitive subset of SGD mutants directly within cultured phagocytes by monitoring replication of these 125 strains over 7 h of growth in RAW264.7 macrophages. For reference, we included several mutants with known intracellular replication defects (Supplementary Data 2). Surprisingly, we observed intracellular growth defects for only 62 LPM-sensitive mutants, while the remaining 63 genes were dispensable for intramacrophage growth (Fig. 1b, Supplementary Data 2). From this, we

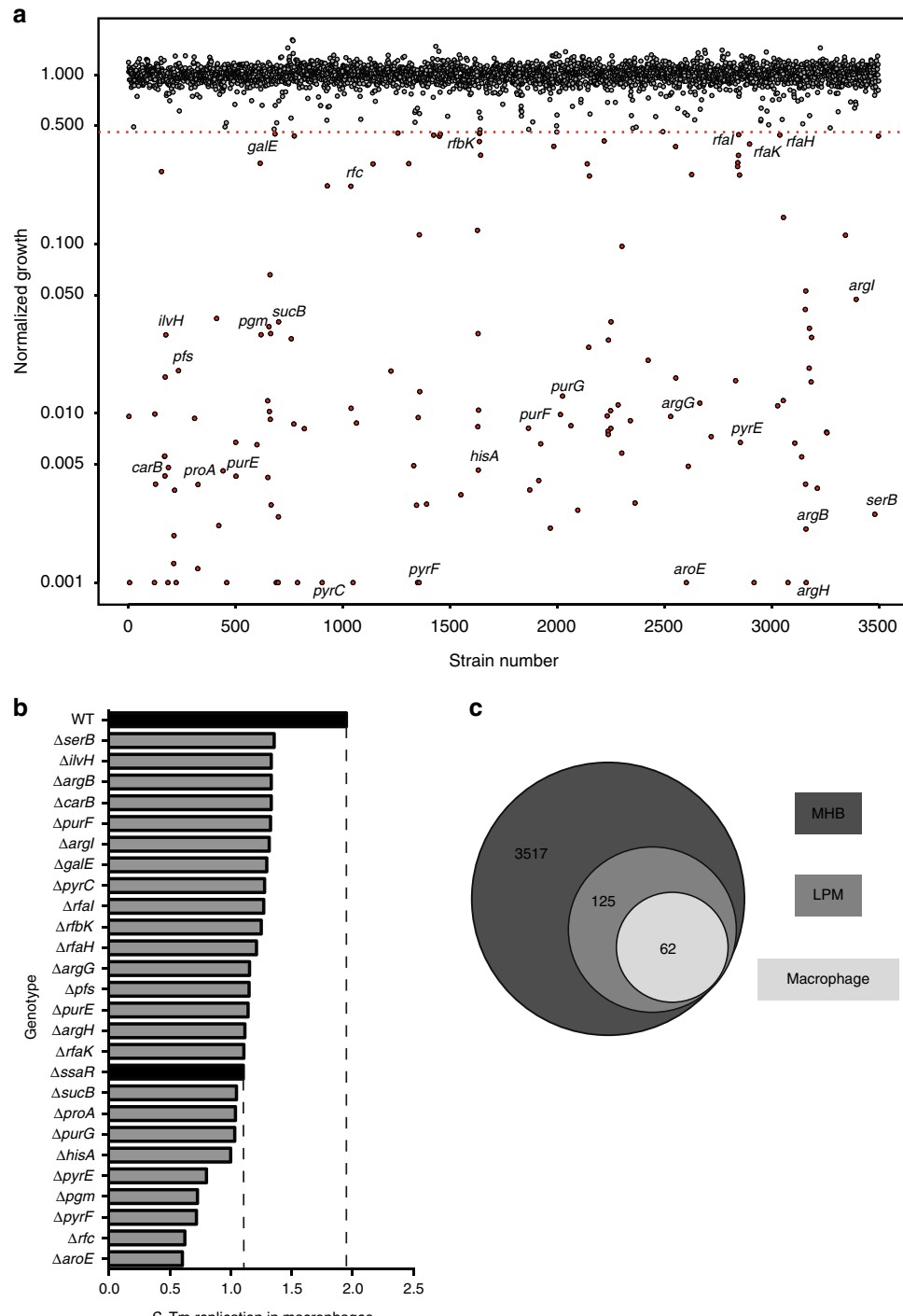

**Fig. 1** Genetic requirements for *S.* Tm growth in host-mimicking media and macrophages. **a** Index plot showing normalized growth of mutant strains from the *Salmonella* single-gene deletion (SGD) collection in LPM media, sorted in order of chromosomal position of deleted genes. Values shown per strain represent the calculated mean growth of three replicate screens, normalized to account for plate and positional effects. Points below the red dotted line represent genes with growth values less than 3.5 s.d. from the mean of the dataset. Strains that exhibited low growth and were used in follow-up experiments are labeled. **b** Replication of selected mutant strains from the SGD collection in RAW264.7 macrophages over 7 h. Wildtype (WT) and ΔssaR strains (black bars) were used as controls for high and low replication, respectively. Bar plots depict the mean fold-change in bacterial burden between 0 and 7 h of intracellular infection, measured from two technical replicates. **c** Cartoon representing the overlap between genes essential for growth in LPM or within RAW264.7 macrophages, and dispensable genes in *S.* Tm represented in the SGD

inferred that macrophages may be more permissive for bacterial growth relative to LPM. However, we note that our preliminary LPM screen expanded the target space for intracellular-selective antimicrobials beyond growth in conventional media (Fig. 1c).

**Identification of chemical inhibitors of intracellular *S.* Tm.** To identify putative antimicrobials effective against intracellular *S.* Tm, we conducted two parallel chemical screens with an annotated compound collection composed largely of previously

approved drugs (Supplementary Table 1). We screened *S*. Tm grown in (i) LPM and (ii) RAW264.7 macrophages. The collection of 1600 chemicals used in these screens includes ~250 known antibacterial compounds with defined targets in Gram-positive and Gram-negative bacteria. Considering the discordance observed with the growth phenotypes in LPM and macrophages, for some of the SGD strains, we elected to conduct chemical screens in both of these conditions, to maximize our potential to access novel target space (Fig. 2a). To perform the intracellular screen, we measured luminescence from a constitutively expressed luciferase reporter in *S*. Tm to enable in situ estimations of intramacrophage bacterial viability. Luminescence correlated linearly with the number of viable bacteria, and the presence of this reporter construct did not affect bacterial replication within macrophages (Supplementary Fig. 1). Together, these high-throughput screens identified 130 compounds with growth-inhibitory activity (Supplementary Data 3). Notably, 63% of compounds active in LPM were also effective at limiting growth of macrophage-internalized bacteria. However, 54% of compounds active against intracellular *S*. Tm displayed no activity against *S*. Tm grown in LPM (Fig. 2a). The latter class of compounds might exert antibacterial activity through targeting of macrophage-encoded proteins or a bacterial target that is not required for growth in LPM.

We next analyzed the potency of all 130 actives from our primary screens. Conditions included *S*. Tm grown in LPM, RAW264.7 macrophages, as well as standard nutrient-rich (cation-adjusted Mueller-Hinton Broth, MHB) and nutrient-poor (MOPS glucose minimal media) microbiological media. We also measured lactate dehydrogenase (LDH) release from uninfected macrophages as a measure of toxicity and did not pursue compounds with toxicity > 25% (percentage relative to maximum LDH release, see Materials and Methods) (Fig. 2b, Supplemental Data 4). Using an arbitrary cutoff for minimum inhibitory concentration (MIC) of 6.25 μM, we eliminated compounds with activity in MHB, as these were least likely to be intramacrophage-selective in vivo. We found that nucleoside analogs (e.g., doxifluridine, fluorouracil, azacitidine, carmofur) were effective against intracellular *S*. Tm and bacteria grown in LPM or MOPS minimal media, but not MHB. As genes involved in nucleotide biosynthesis were required for intracellular growth (Fig. 1, e.g., *pyrC*, *pryE*, *pyrF*, *purE*, *purF*, *purG*), these data suggest that our genetic and chemical screens accurately probed the intramacrophage environment encountered by *S*. Tm. Further, when considering all compounds with an MIC ≤ 3 μM against intracellular *S*. Tm, 64% of these compounds were similarly potent in LPM compared to only 50% in MHB or MOPS (Supplemental Data 5). We prioritized 15 of the 31 compounds that were exclusively active in LPM/macrophages and not MHB/MOPS, based on low host cell cytotoxicity, potency, chemical diversity, and commercial availability.

We tested the 15 priority actives for the ability to reduce intracellular replication (measured by CFU enumeration) of *S*. Tm in primary bone marrow-derived macrophages isolated from C57BL/6 mice (Fig. 2c). Ciprofloxacin is an antibiotic routinely used in salmonellosis infection treatment and was therefore included as a positive control. Of the 15 compounds tested, 11 significantly reduced bacterial replication within 4 h at a concentration of 128 μg mL$^{-1}$ ($P < 0.05$, two-way ANOVA, Bonferroni multiple test correction). Excluding those with previously characterized antibacterial activity, 4 compounds reduced bacterial viability > 2-fold at a concentration of 64 μg mL$^{-1}$ or less: bromperidol, metergoline, ciclopirox, and ethopropazine. The four remaining compounds had moderately increased activity in LPM relative to MHB (Fig. 2d), suggesting selectivity of these antimicrobials for an in vivo-mimicking

environment. Metergoline was the most potent of these compounds and was the focus of subsequent studies of mechanism and in vivo efficacy.

**The outer membrane antagonizes metergoline activity**. The MIC of metergoline in primary macrophages is significantly lower (~8 μg mL$^{-1}$) than in LPM (128 μg mL$^{-1}$) or MHB (>256 μg mL$^{-1}$). These discrepancies led us to hypothesize that one or more aspects of the SCV induce metergoline hypersensitivity in *S*. Tm. To identify potential contributors to this conditional susceptibility, we sought to identify conditions that would increase metergoline activity in standard growth media (MHB). We therefore conducted a chemical-genetic screen of the SGD collection in the presence of a sub-lethal concentration of metergoline in MHB (100 μg mL$^{-1}$) and compared the growth of each mutant to an untreated control (Fig. 3a and Supplementary Data 6). In this experiment, gene deletions that resulted in metergoline hypersensitivity could provide insight to bacterial processes that confer metergoline resistance in nutrient-rich growth media (MHB).

Our chemical-genetic data revealed that genes involved in assembly of LPS (*rfaG*, *rfaQ*), outer membrane (OM) integrity (*asmA*, *tolQ*, *tolR*, *yfgL*), synthesis and turnover of cell wall (*ldcA*, *prc*, *nlpI*), or RND efflux pumps (*tolC*, *acrB*) are required for normal growth in the presence of metergoline. Indeed, we confirmed that deletions of *tolC*, *tolR*, and *rfaQ* in *S*. Tm all resulted in hypersensitivity to metergoline in the intracellular environment (Fig. 3b). From these data we inferred that (i) disruption of cell envelope integrity increases metergoline potency, (ii) metergoline is a likely substrate of the AcrAB-TolC efflux pump, and (iii) increased OM permeability and/or decreased efflux-pump activity might occur in LPM/macrophages to permit metergoline activity.

We noted in our chemical screens that several antibiotics with poor activity against Gram-negative bacteria (e.g., telithromycin, erythromycin, piperacillin, meropenem, rifampin, rifaximin, mupirocin, novobiocin) inhibited growth of *S*. Tm in LPM and/or macrophages. The activity of these antibiotics against Gram-negative bacteria is enhanced by OM permeabilization[32,33]. Considering the hypersensitivity of OM mutants to metergoline, we reasoned that OM-perturbing agents would synergize with metergoline. In line with this, we observed synergy between metergoline and several OM-perturbing agents, including EDTA, polymyxin B, and polymyxin B nonapeptide (Fig. 3c). We also speculated that metergoline potency in macrophages could be driven by the presence of bicarbonate as a buffer in tissue culture media, as we and others have previously reported bicarbonate-mediated potentiation of multiple antibiotic classes[21,34]. Although physiological concentrations of bicarbonate (25 mM) alone had no effect on metergoline's activity against *S*. Tm grown in MHB, it did enhance its antibacterial activity when combined with polymyxin B (Fig. 3c).

Our chemical-genetic screen data also suggested a potential role for efflux in conferring metergoline resistance in MHB. Indeed, we observed a ≥ 8-fold increase in potency of metergoline against a Δ*tolC* strain of *S*. Tm grown in MHB, and polymyxin B potentiated metergoline against an efflux-deficient strain of *S*. Tm (Fig. 3d). However, others have shown that TolC is required, and therefore active, for intracellular growth of *S*. Tm[35,36]. We therefore reasoned that OM permeability, and not reduced efflux, contributes to metergoline activity in macrophages.

Growth of *S*. Tm in LPM results in a ≥4-fold increase in sensitivity to metergoline compared to growth in MHB. In this media, *S*. Tm is resistant to polymyxin B; accordingly, it did not synergize with metergoline (Supplementary Fig. 2A). However, a

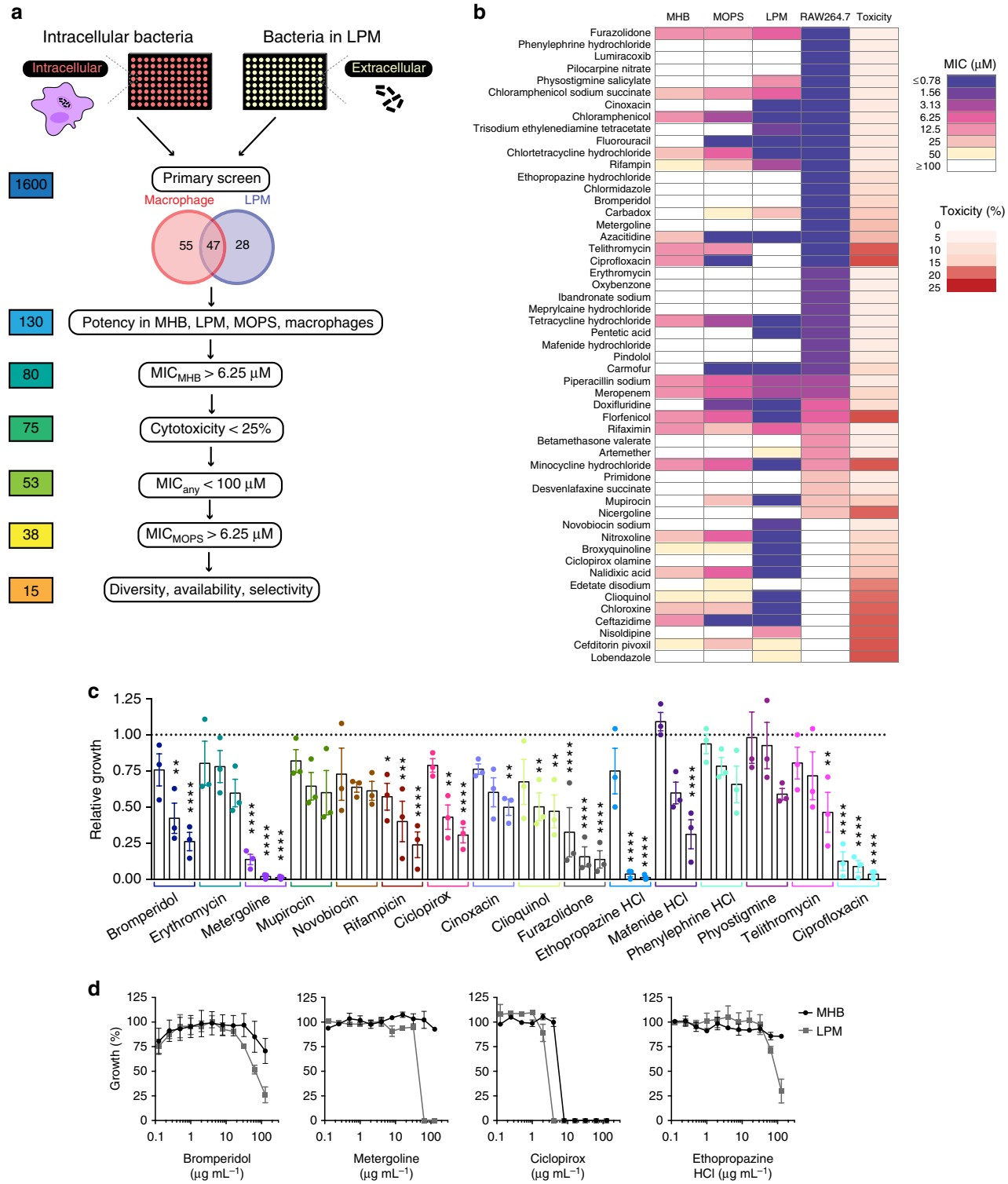

**Fig. 2** Chemical screen identifies novel compound activities against intracellular *S*. Tm. **a** Screening workflow to identify compounds with intracellular antimicrobial activity against *S*. Tm grown in acidic LPM media and internalized in RAW264.7 macrophages. Secondary screening pipeline is shown below with the number of compounds remaining at each step to the left. **b** Potency and toxicity analysis of all primary screen actives represented as a heat map. Shown are the minimum inhibitory concentrations (MIC) for all compounds against *S*. Tm grown in MHB, LPM, MOPS (OD$_{600}$), and inside RAW264.7 macrophages (luminescence, RLU); the final column (toxicity) reports lactate dehydrogenase release from RAW264.7 macrophages after 2 h of exposure to 50 μM compound. All values shown reflect the mean of duplicate measurements. **c** Intracellular *S*. Tm replication measured in primary bone marrow-derived macrophages (BMMs) isolated from C57BL/6 mice. Relative growth reflects replication over 4 h, normalized to bacterial growth in BMMs treated with DMSO. Compounds were added at 8, 64, and 128 μg mL$^{-1}$, as shown in increasing concentrations for each. Bar plots depict the mean of three independent biological replicates, error bars indicate s.e.m. Groups were compared via two-way ANOVA with Bonferroni correction for multiple testing. *$P < 0.05$, **$P < 0.01$, ***$P < 0.001$, ****$P < 0.0001$. **d** Potency analysis of *S*. Tm growth inhibition for bromperidol, metergoline, ciclopirox, ethopropazine HCl in MHB (black) and LPM (gray). Growth is normalized to a DMSO control (set to 100%), error bars indicate s.e.m. for two biological replicates

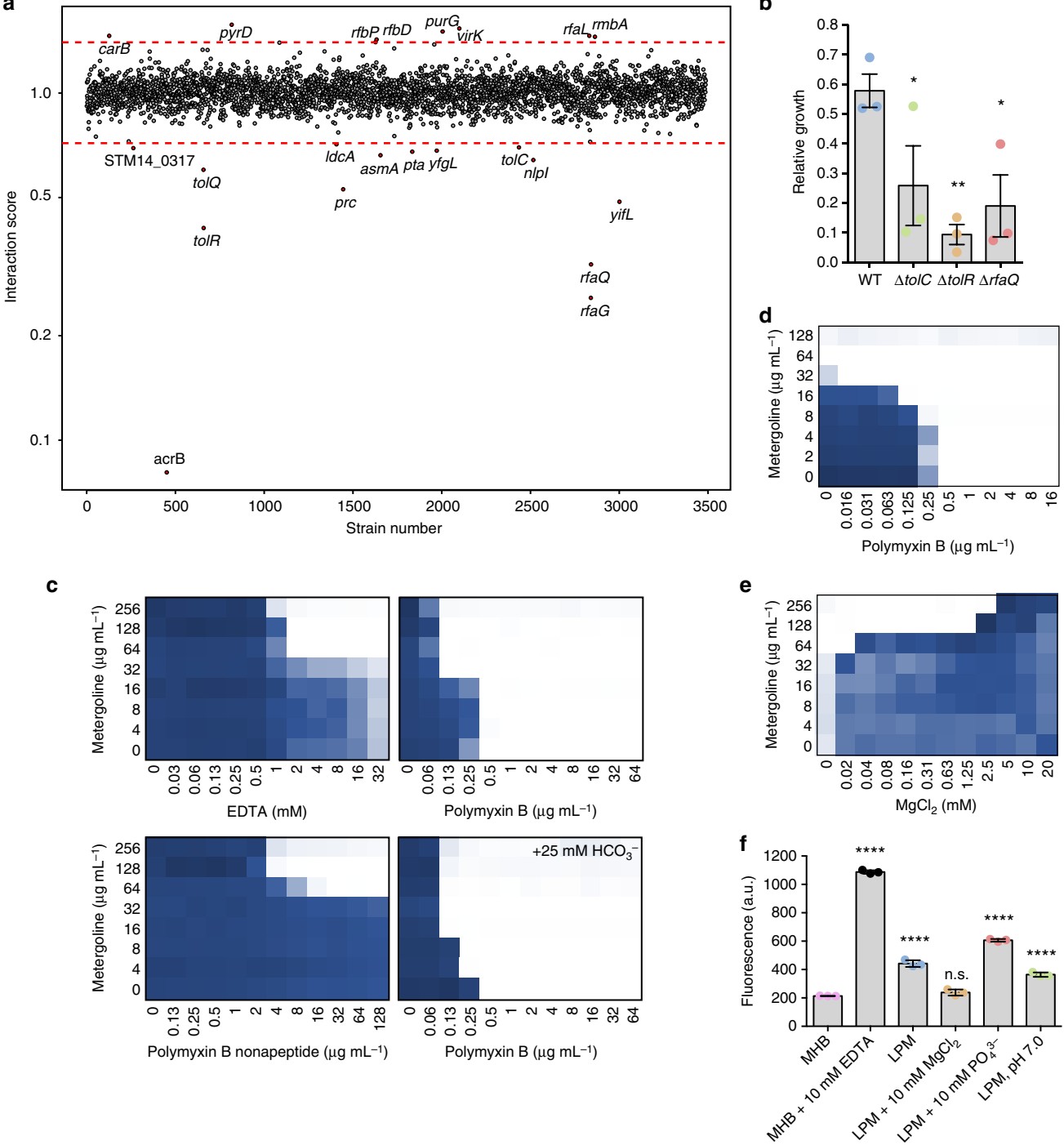

**Fig. 3** Impact of OM integrity and efflux on metergoline activity in MHB and LPM. **a** Index plot showing sensitivity of SGD collection mutant strains grown in MHB with 100 µg mL$^{-1}$ metergoline. Strains are sorted based on chromosomal position of the deleted gene. The chemical-genetic interaction score was calculated by dividing normalized growth of each mutant in the presence of metergoline divided by normalized growth in MHB. Red lines indicate 3 s.d. from the mean of the dataset and values represent the mean of duplicate screens. The deleted genes within sensitive (below red) or resistant (above red) mutant strains are indicated. **b** Intracellular *S*. Tm (WT str. 14028s and indicated SGD mutants) replication measured in bone marrow-derived macrophages isolated from C57BL/6 mice, treated with 8 µg mL$^{-1}$ metergoline. Relative growth reflects replication over 4 h, normalized to bacterial growth in macrophages treated with DMSO. Bar plots depict the mean of three independent infections, error bars indicate s.e.m. SGD mutant groups were compared to WT *S*. Tm with a Kruskal–Wallis test with Dunn's multiple test correction. *$P < 0.05$, **$P < 0.01$. **c** Chequerboard broth microdilution assay showing dose-dependent potentiation of metergoline by membrane-perturbing agents against *S*. Tm grown in MHB. Where indicated, sodium bicarbonate (HCO$_3^-$) was added to media at a final concentration of 25 mM. **d** As in **c** but with a Δ*tolC* strain of *S*. Tm. **e** Chequerboard assay showing antagonism between Mg$^{2+}$ and metergoline in LPM. In **c–e**, higher growth is indicated in dark blue and no detectable growth in white. Results are representative of at least two independent experiments. **f** NPN uptake assay for WT *S*. Tm grown in variants of LPM, MHB, or MHB with 10 mM EDTA. Values were normalized to account for background fluorescence prior to plotting. Bar plots depict the mean of triplicate experiments, error bars indicate s. d. All groups were compared to MHB via one-way ANOVA with Holm–Sidak's multiple test correction. ****$P < 0.0001$

$\Delta phoP$ strain of *S.* Tm was sensitized to polymyxin B, and in this strain we observed strong synergy between polymyxin B and metergoline in both LPM and MHB (Supplementary Fig. 2A, Supplementary Fig. 2B).

We next sought to determine the component(s) of LPM that contribute to increased OM permeability. In addition to lower pH, LPM has decreased $Mg^{2+}$ and $PO_4^{3-}$ concentrations relative to MHB, reflective of the environment in the SCV. Since WT *S.* Tm was sensitized to metergoline, rifampicin, and mupirocin when grown in LPM, we measured the MIC of these compounds against bacteria grown in ion-supplemented (10 mM $MgCl_2$ or $PO_4^{3-}$) or pH-adjusted (pH 7.0) media. Additionally, we measured the activity of vancomycin in these media because similar to rifampicin and mupirocin, vancomycin has low activity against Gram-negative bacteria due to the OM permeability barrier[32]. For all four compounds tested, addition of 10 mM $MgCl_2$ to LPM suppressed activity by at least four-fold. The effect of increased pH or $PO_4^{3-}$ supplementation was compound-dependent; for metergoline, only $Mg^{2+}$ supplementation suppressed activity (Supplementary Fig. 2C).

We observed antagonism between metergoline and $Mg^{2+}$ in LPM, which we attribute to stabilization of the OM by $Mg^{2+}$ and a corresponding decrease in metergoline penetration (Fig. 3e). Accordingly, we observed > 2-fold enhancement of *N*-phenyl-1-naphthylamine (NPN) uptake for bacterial cells grown in LPM relative to MHB, which was abrogated by supplementation with 10 mM $MgCl_2$ (Fig. 3f). Measurement of NPN uptake offers a direct measurement of OM integrity, as an intact OM prevents entry of this hydrophobic fluorophore into the phospholipid bilayer where NPN exhibits the highest fluorescence[37]. As expected, EDTA significantly increased NPN uptake of bacterial cells in MHB, consistent with the difference in metergoline MIC we observe between LPM and MHB supplemented with EDTA (Fig. 3c, f).

These data suggested that the antibacterial activity of metergoline was dependent on OM permeabilization and driven by $Mg^{2+}$ limitation in LPM/macrophages, leading us to hypothesize that the OM of *S.* Tm is perturbed in the intracellular environment. We and others have shown that OM weakening by metal depletion or cationic molecules results in increased surface roughness and the appearance of pits in the OM[38-40]. To study these features of OM disruption in intracellular *S.* Tm, we used atomic force microscopy (AFM) to measure surface roughness of intracellular bacteria and compared this to bacteria grown in MHB or LPM. Cells grown in MHB had a largely uniform cell surface with a maximum peak to pit roughness of 9.37 nm. Remarkably, we observed deep pits on the surface of cells grown in LPM with a maximum roughness score of 22.6 nm (Fig. 4a, b and Supplemental Data 7). We observed an almost 50% increase in surface roughness (13.7 vs 9.52 nm) for *S.* Tm directly isolated from primary bone marrow-derived macrophages (see Methods) relative to cells grown in MHB (Fig. 4c and Supplemental Data 7).

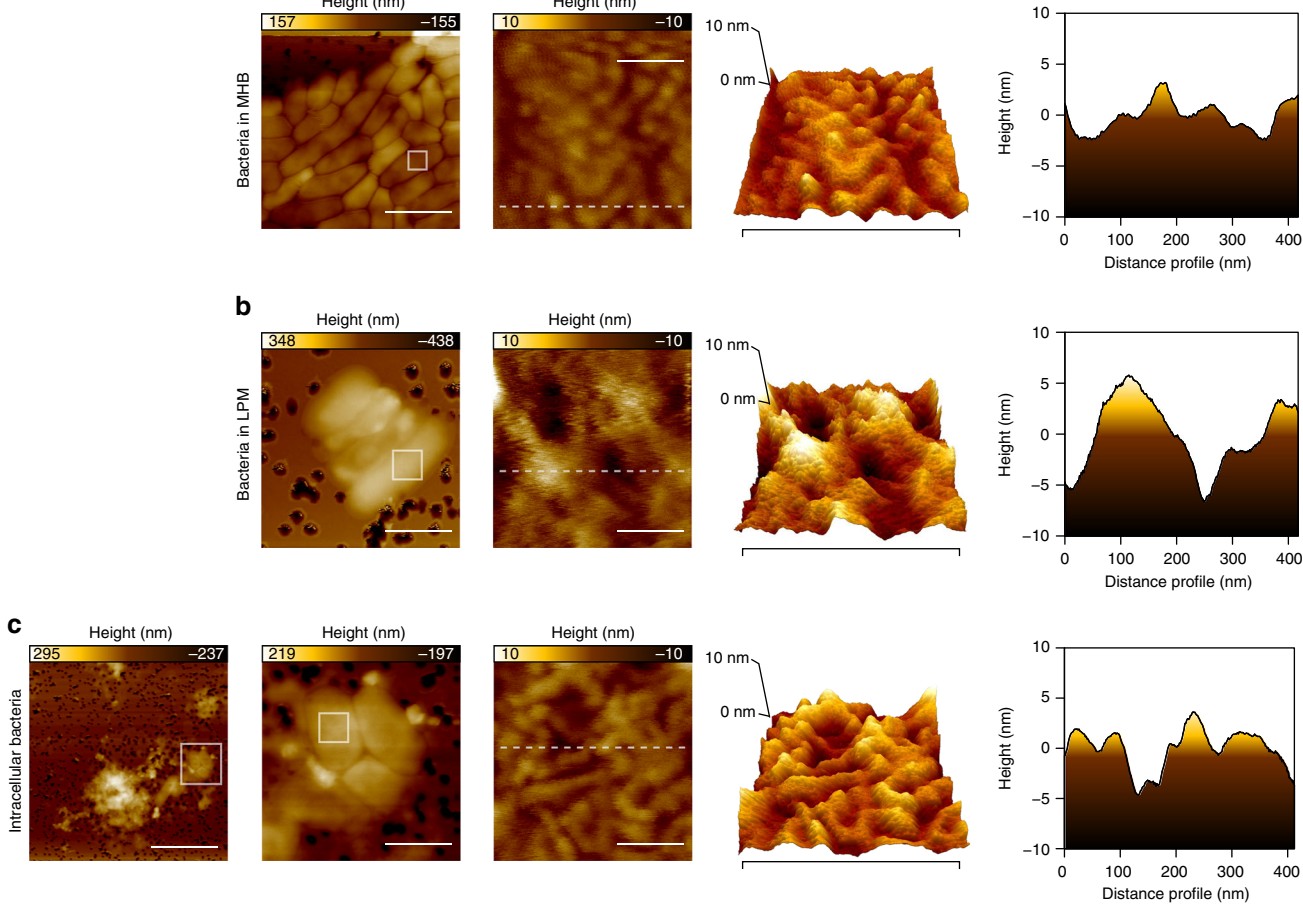

**Fig. 4** Atomic force microscopy (AFM) on *S.* Tm grown in MHB, LPM, or bone marrow-derived macrophages. Images taken at increasing resolution as well as a 3D surface projection are shown for bacteria grown in **a** MHB, **b** LPM, or **c** immediately following lysis of infected macrophages (see Methods). Two-dimensional surface roughness projections (far right) show the surface topology of each sample. Scale bars from left to right show the following distance: **a** 2, 0.1, 0.4 μM; **b** 1, 0.1, 0.4 μM; **c** 4, 1, 0.1, 0.4 μM

Given that OM integrity was protective against metergoline, we reasoned that metergoline might be readily active against Gram-positive bacteria, which do not possess an OM barrier. Indeed, metergoline was ≥16-fold more potent against methicillin-resistant *Staphylococcus aureus* (MRSA) than *S.* Tm in MHB (MIC = 32 μg mL⁻¹, Supplementary Fig. 3A), but roughly equipotent against both bacteria in primary macrophages (Fig. 2c and Supplementary Fig. 3B). Again, these observations are consistent with increased permeability of the Gram-negative OM in the intracellular environment.

**Metergoline perturbs the bacterial cytoplasmic membrane.** Sensitivity to metergoline could be induced in vitro through genetic disruption of efflux. We elected to use a $\Delta tolC$ strain of *S.* Tm to gain insight into metergoline's mechanism of action. To identify a possible protein target for metergoline, we sought to isolate spontaneously resistant mutants by plating at 4x MIC on MHB or LPM agar, but these attempts were unsuccessful. Moreover, we did not detect a significant increase in MIC during a 10-day serial passage experiment with MRSA or $\Delta tolC$ *S.* Tm (Supplementary Fig. 4A). We were, however, able to observe rapid bacteriolytic activity of metergoline against $\Delta tolC$ *S.* Tm (Fig. 5a). This led us to speculate that metergoline conditionally targets the bacterial cytoplasmic membrane in outer membrane-disrupted cells, in line with previous observations of antibiotic-induced bacterial lysis attributed to imbalanced proton homeostasis across this membrane[41,42]. This hypothesis would be consistent with the cryptic antifungal activity of metergoline that is related, in part, to depolarization of mitochondrial membrane potential[43].

In Gram-negative bacteria, the inner cytoplasmic membrane regulates the proton motive force (PMF) to generate energy that is necessary for ATP synthesis by the $F_1-F_0$ ATPase[44]. The PMF is composed of electrical potential ($\Delta\psi$) and a transmembrane proton gradient ($\Delta pH$), and perturbations to either component result in precise compensatory increases to the other[45]. This process may be targeted by membrane potential-uncoupling antibiotics, wherein the dissipation of either $\Delta\psi$ or $\Delta pH$ results in a collapse of the PMF[34,46,47]. Remarkably, we found that metergoline caused a rapid release of 3,3′-dipropylthiadicarbo-cyanine iodide (DiSC₃(5)) (Fig. 5b), a fluorescent probe that accumulates in the cytoplasmic membrane in a $\Delta\psi$-dependent manner[48]. Metergoline caused a similar DiSC₃(5) response to the ionophore valinomycin, a known dissipator of $\Delta\psi$ (Supplementary Fig. 4B). In contrast, the $\Delta pH$ dissipator carbonyl cyanide m-chlorophenyl hydrazone (CCCP) decreases fluorescence of DiSC₃(5), due to a compensatory increase in $\Delta\psi$ (Supplementary Fig. 4C). These data suggest that metergoline treatment rapidly decreases electrical potential at the cytoplasmic membrane. In line with this, metergoline synergized with CCCP against WT *S.* Tm grown in MHB supplemented with EDTA, or LPM (Fig. 5c), consistent with previous observations of antibacterial synergy between dissipators of $\Delta pH$ and $\Delta\psi$[47].

Disruption of the cytoplasmic membrane potential by metergoline would be expected to perturb cellular ATP levels. Indeed, we found that cellular ATP levels were reduced ~10-fold after a 30 min exposure to 128 μg mL⁻¹ metergoline (Fig. 5d). By comparison, CCCP at a concentration of 32 μg mL⁻¹ led to a <2-fold-change in ATP levels in this short experiment (Supplementary Fig. 4D). Lastly, we observed a similar effect of

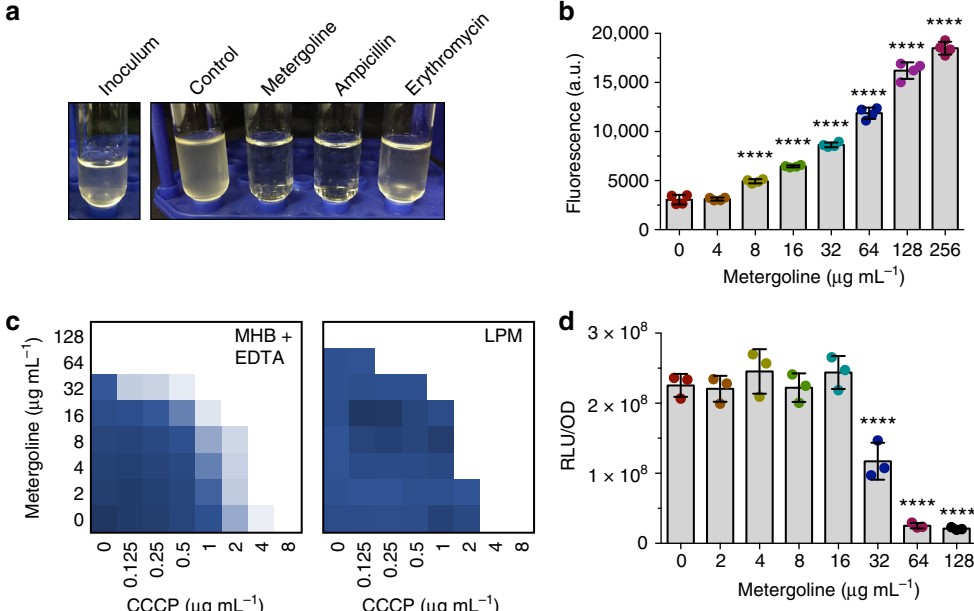

**Fig. 5** Metergoline is bacteriolytic and perturbs the *S.* Tm cytoplasmic membrane. **a** Turbidity of cultures of *S.* Tm $\Delta tolC$ in MHB after growth to mid-log phase (left, inoculum) then 2.5 h of growth at 37 °C in the presence of metergoline (200 μg mL⁻¹), ampicillin (16 μg mL⁻¹), erythromycin (16 μg mL⁻¹), or a DMSO control. Note that erythromycin is bacteriostatic and culture turbidity did not change relative to the inoculum; ampicillin (bactericidal) and metergoline both cleared culture turbidity. **b** DiSC₃(5) assay on late-log phase *S.* Tm grown in MHB supplemented with 10 mM EDTA to enable DiSC₃(5) binding to the cytoplasmic membrane. Cells were loaded with DiSC₃(5) prior to a 1 min incubation with increasing concentrations of metergoline. Bar plots depict the mean of two biological replicates, error bars indicate s.d. All groups were compared against 0 μg mL⁻¹ metergoline via one-way ANOVA with Holm–Sidak's multiple test correction. ****$P$ < 0.0001. **c** Chequerboard broth microdilution assay showing synergy between metergoline and CCCP against *S.* Tm grown in MHB with 10 mM EDTA or LPM. **d** *S.* Tm grown in MHB with 1 mM EDTA to early-log phase, then exposed to metergoline for 30 min. Cellular ATP levels were estimated by luciferase activity (relative light units, RLU) normalized to optical density (OD₆₀₀). Bar plots depict the mean of two biological replicates, error bars indicate s.d. Groups were compared against 0 μg mL⁻¹ metergoline via one-way ANOVA with Holm–Sidak's multiple test correction. ****$P$ < 0.0001

metergoline on DiSC$_3$(5) release in MRSA (Supplementary Fig. 3C), as well as synergy between CCCP and metergoline (Supplementary Fig. 3D).

**In vivo efficacy of metergoline in a murine infection model.** Given its selective potency in vitro, we tested the efficacy of metergoline in a systemic murine infection model with *S.* Tm. Genetically susceptible mice (*Nramp*-deficient C57BL/6) intraperitoneally infected with *S.* Tm typically succumb within ~72 h. This route of infection recapitulates severe salmonellosis and bacteraemia is detectable as early as 1 h post injection[49]. Metergoline is a naturally occurring alkaloid compound derived from ergot fungus, and, to our knowledge, its in vivo efficacy has been explored solely in experiments to characterize its anxiolytic effects in mice as a serotonin antagonist[50–52]. We first administered metergoline at multiple doses to uninfected animals to test for potential adverse effects. We observed weight loss and increased agitation of otherwise healthy mice following administration of metergoline twice daily at 10 mg kg$^{-1}$, but more mild effects at 5 mg kg$^{-1}$ (average 6.5% weight loss over ~3 days). We therefore selected 5 mg kg$^{-1}$ as our therapeutic dose for metergoline.

We determined the efficacy of metergoline as a potential therapeutic by testing its ability to reduce bacterial load and prolong survival when administered either at the time of infection (Fig. 6a, b) or 12 h after infection (Supplementary Fig. 5E,F). Metergoline treatments were administered at 5 mg kg$^{-1}$, twice daily, until experimental endpoint was reached for both vehicle-treated and metergoline-treated groups. When administered at the time of infection, metergoline significantly reduced bacterial load in all organs harvested (Fig. 6a) and significantly extended animal survival time (Fig. 6b). We also observed a statistically significant reduction in bacterial counts in the colon of *S.* Tm-infected mice when administering metergoline starting at 12 h after infection, and a reduction in counts in the spleen, liver and cecum, although not statistically significant (Supplementary Fig. 5A). Under this treatment regimen, metergoline also significantly prolonged animal survival (Supplementary Fig. 5B).

**Antibacterial activity of metergoline analogs.** Given the potential to extend animal survival time following frequent administration of metergoline despite mildly adverse effects, we aimed to identify a chemical analog with increased potency against *S.* Tm and/or decreased toxicity against eukaryotic cells. We therefore pursued an in-depth structure–activity-relationship analysis to explore the chemical properties of the metergoline scaffold required for activity. We tested seventeen structurally related analogs for the ability to reduce growth of wildtype and Δ*tolC S.* Tm in MHB, LPM, and macrophages, as well as MRSA grown in MHB (Supplementary Figure 6, Supplementary Table 2). Four of the analogs we evaluated are FDA-approved therapeutics that contain the core ergot alkaloid scaffold but diverge structurally from metergoline in multiple regions. These compounds: nicergoline, pergolide, cabergoline, and methysergide, are neuroactive drugs that interact with dopamine and/or serotonin receptors, similar to metergoline. Of these, nicergoline showed modest antimicrobial efficacy while the remaining three drugs were inactive in our assays. Thirteen additional analogs were synthesized from metergoline by systematic replacement of the benzyl carbamate moiety with other groups (Supplementary Figures 7-20, Supplementary Table 3; compounds **S1-S11**, **MLBC-01** and **MLBC-02**).

Given the mildly acidic pH of LPM and SCV, we considered the possibility that metergoline may be acting as a pro-drug by means of hydrolysis of the acid-labile carbamate moiety to release its corresponding amine (Supplementary Fig. 6). Evaluation of compound S1, however, showed no in vitro efficacy at 128 μg mL$^{-1}$ in all but one assay (MHB supplemented with EDTA) and inferior intramacrophage growth inhibition relative to metergoline. The remaining synthetic analogs consisted of two alkyl carbamates, seven amide derivatives, a benzyl urea derivative, a sulfonamide, and an amine. Most of these analogs were inactive in our assays (Supplementary Table 2) but the α,β-unsaturated amides **MLBC-01** and **MLBC-02** demonstrated antimicrobial activity superior to metergoline (Table 1). In vitro, **MLBC-01** was two to four times as potent as metergoline while the chlorophenyl derivative **MLBC-02** was up to 64-fold more potent (against MRSA). Both compounds also inhibited intramacrophage growth of *S.* Tm with similar potency to metergoline. However, compounds **MLBC-01** and **MLBC-02** were more toxic to macrophages (18.1% and 8.9% toxicity, respectively) relative to metergoline (2.9% toxicity) as determined by lactate

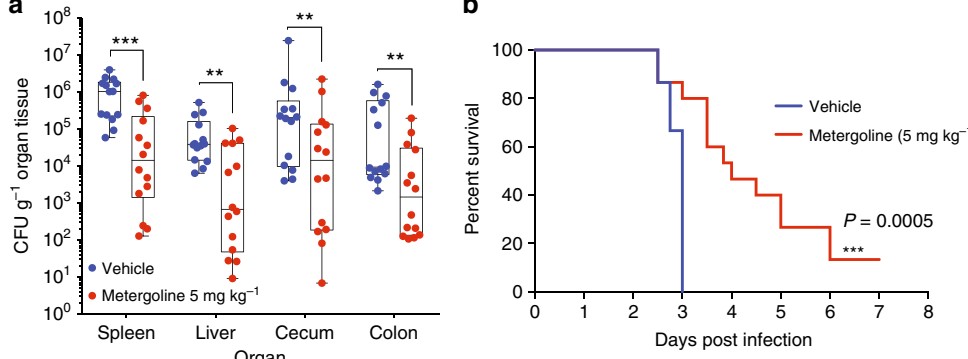

**Fig. 6** In vivo efficacy of metergoline in a murine model of systemic *S.* Tm infection. C57BL/6 mice were infected intraperitoneally (i.p.) with ~10$^5$ CFU *S.* Tm. **a** Groups of mice were treated twice daily (every 12 h) with metergoline (5 mg kg$^{-1}$, red bars) or DMSO (5% in DMEM, blue bars) by i.p. injection. Treatments were administered beginning at the time of infection. Mice were euthanized at experimental endpoint (60 h post infection). Bacterial load in the spleen, liver, cecum, and colon was determined by selective plating on streptomycin. Data shown are the means of three separate experiments (*n* = 5 per group). Box plot whiskers show the minimum to maximum values per group, lines in box plots show the median of each group. Groups were analyzed with a two-way ANOVA and corrected for multiple comparisons with a Holm–Sidak test. **b** For survival experiments, groups of mice were treated twice daily (every 12 h) beginning at the time of infection with metergoline (5 mg kg$^{-1}$, red) or DMSO (5% in DMEM, blue) by i.p. injection, and were euthanized at clinical endpoint. Survival curves shown are from three separate experiments (*n* = 5 per group). Groups were analyzed with a Gehan–Breslow–Wilcoxon test for survival curve differences. **P<0.01, ***P<0.001

**Table 1 Structure–activity relationship analysis of metergoline and two synthetic analogs**

| Compound[a] | Minimum Inhibitory Concentration (µg mL⁻¹) | | | | | RAW264.7 | |
| | S. Tm WT | | | ΔtolC | MRSA | Growth[c] | Toxicity (%)[d] |
| | MHB | EDTA[b] | LPM | MHB | MHB | | |
| metergoline | >128 | 64 | 128 | 32 | 32 | 0.08 | 2.9 |
| MLBC-01 | >128 | 32 | 128 | 8 | 16 | 0.07 | 18.07 |
| MLBC-02 | >128 | 8 | 32 | 4 | 0.5 | 0.08 | 8.9 |

All numbers reflect the mean from two (MIC) or three (RAW264.7) experiments
[a]Blue highlights in chemical compounds depict structural differences between metergoline and MLBC-01 and MLBC-02
[b]WT S. Tm was grown in MHB with 10 mM EDTA
[c] Intracellular activity was measured by addition of 8 µg mL⁻¹ compound to RAW264.7 macrophages infected with WT S. Tm and is reported as fold-growth inhibition relative to a DMSO-treated control
[d]Toxicity was estimated by lactate dehydrogenase release from uninfected RAW264.7 cells

dehydrogenase release (Table 1). We are encouraged by the improved activity of these analogs against bacteria and further structural optimization of this novel scaffold is presently underway.

## Discussion

Intracellular pathogens often occupy niches that are difficult to recapitulate in vitro; thus, we and others have developed fluorescence, luminescence, and colorimetric-based screening assays with readouts that approximate bacterial viability when compared to traditional cfu counting methods[36,53–56]. Here we describe a macrophage-based chemical screen specific for intracellular S. Tm, leading to the discovery of metergoline as an antibacterial with in vivo efficacy, and the unexpected identification of outer membrane disruption in the intracellular environment. Further, we show that metergoline disrupts the proton motive force across the S. Tm cytoplasmic membrane, diminishes cellular ATP levels, reduces bacterial load in a murine model of systemic S. Tm infection, and significantly prolongs survival of infected animals. Interestingly, we found that metergoline activity in vivo was particularly potent in the cecum and colon, which is perhaps surprising given that the spleen and liver serve as the primary reservoirs for intraperitoneal S. Tm infection. Because the presence of bacteria in the intestinal tract after intraperitoneal infection is primarily due to transit from the hepatic biliary system[57], we speculate that this phenotype merely reflects the lower bacterial burden in the liver, comprising a smaller reseeding population. Additionally, though we identify a disruption of the proton motive force and reduction in cellular ATP levels concomitant with metergoline treatment, we note that these effects may be indirect consequences of bacterial cell death. Indeed, future work remains to completely elucidate the mechanisms underlying metergoline activity.

Antimicrobials effective against intracellular pathogens are ideally multifaceted and with the following properties:

(i) penetration of host cell membranes, (ii) penetration of bacterial membrane(s), (iii) the ability to perturb a conditionally essential bacterial target required for intracellular survival, and (iv) bacterial target specificity to limit host cell toxicity. Primary chemical screens directly against bacteria internalized within host cells are essential to identify molecules with all of these properties; however, these screens are technically challenging and poorly suited to high-throughput applications. Here, we adopted an alternative approach of screening in host-mimicking media (LPM) to perturb conditional bacterial targets (i.e., nutrient biosynthesis and cell envelope homeostasis) in the intracellular environment. Although we acknowledge that our genetic and chemical screen data suggest that LPM is an imperfect mimic of macrophages, we were indeed able to identify intracellular-selective antimicrobials that are otherwise elusive using conventional screens in nutrient-rich growth media. Given that the intramacrophage screening platform described here is limited to approximately 1000 compounds per day, the alternative approach using unconventional growth media, such as LPM, is a preferable option for larger-scale chemical screening efforts.

We showed that $Mg^{2+}$ depletion increased OM permeability and sensitized S. Tm to metergoline. OM disruption on depletion of $Mg^{2+}$ is thought to arise because negatively charged neighboring LPS molecules are stabilized through the binding of divalent cations[58]. The low $Mg^{2+}$ concentrations in LPM/macrophages offers a compelling explanation for the intracellular activity of metergoline. Considered with our observation of atypical cell surface characteristics of intracellular S. Tm, we speculate that the Gram-negative OM is disrupted in the intracellular environment. This is somewhat paradoxical given that the environmental cues in LPM/macrophages activate PhoPQ and PmrAB-regulated lipid A modifications (caused by, e.g., low pH, low $Mg^{2+}/Ca^{2+}$ and antimicrobial peptides)[59], most of which are thought to confer resistance to cation depletion and antimicrobial peptide stress by altering OM charge. Nevertheless, the impact of these modifications on OM permeability has not been clearly

tested in vivo, and further work is required to disentangle the relationship between lipid A modifications and resistance to antibiotics in the intracellular environment.

These findings have important implications for antibiotic drug discovery. By convention, Gram-negative bacteria are susceptible only to antibiotics capable of penetrating the OM (e.g., polymyxins and aminoglycosides) or to small hydrophilic antibiotics that can traverse the OM through porin channels[58,60]. Our work indicates that natural components of the innate immune system may sensitize intracellular Gram-negative bacteria to otherwise poorly penetrating antibacterial compounds in vivo. Moreover, screening for compounds that are selective for bacteria within macrophages can reveal those that target conditionally essential genes, due to the unique conditions of the intracellular environment. Indeed, genes that are essential for growth under conditions that more closely resemble those during infection is a promising avenue in modern antibiotic research that has garnered increased attention in recent years[6]. For complex S. Tm infections that include several bacterial subpopulations with highly varied gene expression and virulence phenotypes, the identification of novel antimicrobials with unique targets is critical. We conclude that intracellular-targeting chemical screening platforms have the potential to identify small molecules with relatively under-explored targets, a goal of paramount importance in the current antibiotic resistance era.

## Methods

**Ethics statement**. All animal experiments were performed according to the Canadian Council on Animal Care guidelines using protocols approved by the Animal Review Ethics Board at McMaster University under Animal Use Protocol #17-03-10.

**Reagents**. Screening stocks (5 mM) of the Pharmakon-1600 (MicroSource) compound library were stored at −20 °C in DMSO. With the following exceptions, all chemicals were purchased from Sigma-Aldrich: Ciclopirox (Santa Cruz Biotechnology), Pergolide (Toronto Research Chemicals), Cabergoline (Toronto Research Chemicals), Methysergide (Toronto Research Chemicals). Compounds were routinely dissolved in DMSO at a concentration of 12.8 or 25.6 mg mL$^{-1}$ and stored at −20 °C. Synthetic analogs of metergoline are described in Supplementary Methods. All synthetic compounds presented in this work were characterized by $^1$H and $^{13}$C NMR (Supplementary Figures 7-20 and Supplementary Table 3).

**Bacterial strains and culture conditions**. All experiments with *Salmonella enterica* ssp. *enterica* ser. Typhimurium (*S.* Tm) were performed with strain SL1344 or derivatives with the exception of the *S.* Tm single-gene deletion collection[22], which was constructed in the related *S.* Tm str. 14028s background. For consistency, in the experiments presented in Figs. 1 and 3b, the wildtype *S.* Tm 14028s strain was used for comparison. For compound screening and secondary assays, *S.* Tm SL1344 was transformed with pGEN-*lux*[61], and SL1344 Δ*tolC* was generated by standard methods[62]. Briefly, the chloramphenicol resistance cassette from pKD3 was amplified by PCR with primers Del TolC F (5′-CAATTATTTTTACAAATTG ATCAGCGCTAAATACTGCTTCACAACAAGGAGTAGGCTGGAGCTGC-3′) and Del TolC R (5′-AGACCTACAAGGGCACAGGTCTGATAAGCGCAGCGCC AGCGAATAACTTACATATGAATATCCTCCTTAGTTCC-3′) and the resulting amplicon was transformed into electro-competent *S.* Tm SL1344 cells harboring the pSim6 plasmid as previously described[63,64]. Successful recombinants were confirmed by colony PCR using primers Chk TolC F (5′-CGACCATCTCCAGCA GCCAC-3′) and Chk TolC R (5′-GAAAAGGCGAGAATGCGGCG-3′). Experiments with *Staphylococcus aureus* used a Canadian isolate (CMRSA10) of community-acquired methicillin-resistant *S. aureus* USA300[65].

Overnight cultures of bacteria were inoculated with a single colony and routinely grown in LB media (10 g L$^{-1}$ NaCl, 10 g L$^{-1}$ Tryptone, 5 g L$^{-1}$ yeast extract) supplemented with antibiotics as appropriate (streptomycin, 150 μg mL$^{-1}$; chloramphenicol, 25 μg mL$^{-1}$; kanamycin, 50 μg mL$^{-1}$; ampicillin, 100 μg mL$^{-1}$). Where indicated, bacteria were subcultured 1:100 and grown to mid-log phase in cation-adjusted MHB (BBL$^{TM}$ Mueller-Hinton II Broth—Cation Adjusted), LPM (5 mM KCl, 7.5 mM (NH$_4$)$_2$SO$_4$, 0.5 mM K$_2$SO$_4$, 10 mM Glucose, 49 μM MgCl$_2$, 337 μM PO$_4$$^{3-}$, 0.05% casamino acids, 80 mM MES, pH 5.8), or MOPS glucose minimal medium (Teknova) supplemented with 40 μg mL$^{-1}$ histidine when growing *S.* Tm SL1344. For LPM supplementation experiments, MgCl$_2$ or KH$_2$PO$_4$ were added to a final concentration of 10 mM and media was pH-adjusted prior to filter sterilizing. Bacteria were grown at 37 °C.

**Genetic screening**. The *Salmonella* single-gene deletion (SGD) library[22] was pinned from frozen DMSO stocks at 384-colony density onto LB agar medium containing 50 μg mL$^{-1}$ kanamycin using a Singer RoToR HDA (Singer Instruments) and grown for 18 h at 37 °C. The SGD was then grown overnight in 384-well clear flat-bottom plates (Corning) in either MOPS glucose (Fig. 1a) or LB (Fig. 3a) supplemented with 50 μg mL$^{-1}$ kanamycin. The Singer RoToR HDA was then used to inoculate assay plates (containing 50 μL per well LPM without casamino acids, MHB, or MHB with 100 μg mL$^{-1}$ metergoline) with a starting inoculum of ~$1.7 \times 10^5$ CFU per well. Optical density at 600 nm (OD$_{600}$) was measured with a Tecan M1000 Infinite Pro plate reader at the time of inoculation ($T_0$) and after 16 h ($T_{16}$) incubation at 37 °C with shaking at 220 rpm. Growth was calculated by subtracting the pre-reads ($T_{16} - T_0$) and interquartile-mean normalization for plate and well effects[66]. The interaction score was calculated by dividing normalized growth in the presence of metergoline by the MHB control. Experiments were performed in duplicate or triplicate as noted.

**Replication of SGD mutants in macrophages**. Strains from the *Salmonella* single-gene deletion library were selected based on prioritization from genetic screening in LPM (growth < 3.5 s.d. from mean of screening data), along with SPI-1, SPI-2, virulence, motility, and regulatory genes, and WT to use as controls. Strains of interest were grown overnight in 96-well clear flat-bottom plates (Corning) in LB in duplicate.

RAW264.7 macrophages were seeded into 96-well plates in DMEM + 10% FBS at ~$10^5$ cells per well and left to adhere for 20–24 h, incubated at 37 °C with 5% CO$_2$. Overnight cultures of bacteria were diluted to obtain an MOI of ~50:1, then opsonized for 30 min in 20% human serum in PBS at 37 °C. Bacteria were added to each well, and plates were spun down at 500×g for 2 min, then incubated for 30 min at 37 °C with 5% CO$_2$. Media was aspirated and replaced with fresh DMEM containing 100 μg mL$^{-1}$ gentamicin to kill extracellular bacteria for 30 min at 37 °C with 5% CO$_2$. For half the plates, RAW264.7 cells were washed once with PBS, then scraped from the wells and lysed in PBS containing 1% (v/v) Triton-X100, 0.1% (w/v) SDS. Bacterial colony-forming units (CFUs) from each well were enumerated by serially diluting in PBS and plating on LB plates ($T_0$ counts). For the other half of the plates, RAW cells were washed once with PBS, then fresh DMEM was added and cells were incubated at 37 °C with 5% CO$_2$. After 7 h, RAW cells were washed once with PBS, then scraped from the wells and lysed in PBS containing 1% (v/v) Triton-X100, 0.1% (w/v) SDS. Bacterial CFU from each well were enumerated by serially diluting in PBS and plating on LB plates ($T_7$ counts). CFU were averaged (two technical replicates per assay plate) and a CFU ratio (CFU at $T_7$ divided by $T_0$) was calculated to represent fold replication over the course of the experiment.

**High-throughput compound screening**. All chemical screening was performed at the Centre for Microbial Chemical Biology (McMaster University). For chemical screening in LPM, an overnight culture of *S.* Tm SL1344 was subcultured 1:100 in LB, grown to an OD$_{600}$ of 0.5, then diluted 40-fold into LPM and grown to an OD$_{600}$ of 0.3 before a final 1:150 dilution into LPM. Bacterial culture (50 μL) was dispensed into 96-well black, clear flat-bottom (Corning) plates and then 50 μL of each compound (diluted to 20 μM in LPM) was added for a final concentration of 10 μM compound and ~$2 \times 10^4$ CFU per well. The OD$_{600}$ was read immediately after compound addition ($T_0$) and then 16 h later ($T_{16}$). Plate and well effects were normalized by interquartile-mean based methods[66] and compounds reducing growth more than 3σ below the mean were considered primary screen actives. Screening was performed in triplicate.

For the screen against intramacrophage *S.* Tm, RAW264.7 macrophages were seeded 16 hrs prior to infection at ~$2 \times 10^5$ cells per well in 96-well black, clear flat-bottom plates (Corning) in DMEM + 10% FBS and were incubated at 37 °C with 5% CO$_2$. *S.* Tm str. SL1344 transformed with pGEN-*lux* was grown overnight in LB with 100 μg mL$^{-1}$ ampicillin, diluted to obtain a multiplicity of infection (MOI) of 100:1, then opsonized for 30 min in 20% human serum (Innovative Research) in PBS at 37 °C. An equal volume of bacteria (100 μL) was added to macrophages and plates were centrifuged at 500×g for 2 min followed by a 30 min incubation at 37 °C with 5% CO$_2$. Media was then aspirated and replaced with fresh DMEM containing 100 μg mL$^{-1}$ gentamicin to eliminate extracellular bacteria, and plates were incubated for 30 min at 37 °C with 5% CO$_2$. Infected cells were washed with PBS prior to addition of 100 μL of DMEM containing compound at 10 μM. Luminescence was read immediately after compound addition ($T_0$) and plates were then incubated for 03 h at 37 °C with 5% CO$_2$. Luminescence was measured a second time after media was replaced with fresh DMEM + 10% FBS ($T_3$) and plates were incubated for a further 3 hr before luminescence was measured a final time ($T_{Final}$).

**Secondary screening of compounds in macrophages**. Overall, 130 priority compounds were selected from chemical screening in LPM and macrophages. Infection assays in RAW264.7 macrophages were performed as described above, with the following modifications: macrophages were seeded at $10^5$ cells per well, macrophages were pretreated with 100 ng mL$^{-1}$ LPS from *Salmonella enterica* serovar Minnesota R595 (Millipore), and an MOI of 50:1 was used. Compounds were serially diluted two-fold starting at 5 mM to achieve a final concentration of

50 μM with 1% DMSO in DMEM. Luminescence was read immediately after compound addition ($T_0$), and plates were incubated for 20 h at 37 °C with 5% $CO_2$ before luminescence was read again ($T_{20}$). Luminescence readings were averaged (two technical replicates per assay plate) and a luminescence ratio (luminescence at $T_{20}$ divided by $T_0$) was calculated to represent fold replication over the course of the experiment. MICs were estimated based on ability to reduce luminescence by at least 10-fold.

**Cytotoxicity assays.** RAW264.7 macrophages were seeded into 96-well plates in DMEM + 10% FBS and 100 ng mL$^{-1}$ LPS from *Salmonella enterica* serovar Minnesota R595 (Millipore) and left to adhere for 20–24 h, incubated at 37 °C with 5% $CO_2$. Compounds were premixed into DMEM at a final concentration of 50 μM with 1% DMSO, then added to wells. After 2 h of compound treatment, the culture supernatant was collected for analysis of lactate dehydrogenase release. Cytotoxicity was quantified colorimetrically with the Pierce LDH cytotoxicity kit where LDH activity is measured by subtracting Absorbance at 490 nm from Absorbance at 680 nm. Lysis control wells were treated with 10× lysis buffer for 1 h. Percent cytotoxicity was calculated with the formula:

$$\frac{LDH_{Compound\ treated} - LDH_{Spontaneous}}{LDH_{Maximum} - LDH_{Spontaneous}} \times 100\%$$

where $LDH_{Spontaneous}$ is the amount of LDH activity in the supernatant of untreated cells and $LDH_{Maximum}$ is the amount of LDH activity in the supernatant of lysis control wells. The LDH activity in cell-free culture medium was subtracted from each value prior to normalization.

**Bone marrow-derived macrophage assays.** Bone marrow-derived macrophages (BMMs) were collected from the femur and tibia of 6–10 week old female C57BL/6 mice (Charles River Laboratories) and differentiated in RPMI (Gibco) + 10% FBS + 10% L-sup (L929 fibroblast conditioned medium) + 100 U penicillin–streptomycin for 7 days at 37 °C and 5% $CO_2$. Differentiated BMMs were seeded 20–24 h prior to infection in 96-well plates at $10^5$ cells per well in RPMI + 10% FBS + 100 ng mL$^{-1}$ *Salmonella enterica* serovar Minnesota R595 (Millipore) and incubated at 37 °C with 5% $CO_2$. Infection assays were performed as described for RAW264.7 macrophages, with the following modifications: bacteria were not opsonized prior to infection, and an MOI of 50:1 was used. Compounds were added to wells as described previously (see Secondary screening of compounds in macrophages).

**Chequerboard analyses and compound potency analysis.** A single colony of freshly streaked bacteria was used to inoculate LB with appropriate antibiotics (150 μg mL$^{-1}$ streptomycin for *S.* Tm str. SL1344 and derivatives, 25 μg mL$^{-1}$ chloramphenicol for SL1344 Δ*tolC* or Δ*phoP*). Overnight cultures were diluted 100-fold into antibiotic-free MHB or LPM, as appropriate, and grown to mid-log phase ($OD_{600} = 0.4$–$0.8$). Subcultures were then diluted to an $OD_{600} = 0.0001$ (~$1 \times 10^5$ cfu mL$^{-1}$) in assay media and 150 μL of this dilution was added to each well of the 96-well assay plate. For chequerboard experiments, an $8 \times 12$ matrix of two compounds was created with two-fold serial dilutions of each compound. Metergoline, CCCP, nigericin, and valinomycin were dissolved in DMSO, and polymyxin B, polymyxin B nonapeptide, and EDTA were dissolved in water. After addition of bacteria, plates were incubated at 37 °C with shaking for 16 h, at which time the $OD_{600}$ was measured. The minimum inhibitory concentrations (MIC) for various compounds were determined using 11 two-fold dilutions and growth was measured after 16 h. The MIC was the concentration that inhibited growth > 95% when compared to the solvent control. In all experiments, DMSO was present at a final concentration < 2% (routinely 1%).

**NPN uptake assay.** Uptake of the lipophilic dye *N*-phenyl-1-naphthylamine (NPN) was measured essentially as previously described[37]. Briefly, overnight cultures of *S.* Tm were diluted 50-fold into LB and grown to mid-log ($OD_{600} = 0.5$), then diluted again 100-fold into variants of LPM, MHB, or MHB with 10 mM EDTA, and grown to late-exponential phase ($OD_{600}$ ~1.2). Cells were harvested by centrifugation, washed in 5 mM HEPES, pH 7.2, then resuspended to a final $OD_{600} = 1.0$. The cell suspension (50 μL), 40 μM NPN (50 μL), and 5 mM HEPES, pH 7.2 (100 μL) were mixed in black clear flat-bottom 96-well plates immediately before measuring fluorescence (excitation, 340 nm; emission 415 nm) in a Tecan M1000 Infinite Pro plate reader. Background fluorescence of each preparation of cells was subtracted along with background fluorescence of NPN in buffer.

**Atomic force microscopy (AFM).** BMMs were differentiated as described above, and then seeded 16 h prior to infection in 6-well plates at $5 \times 10^6$ cells per well in RPMI + 10% FBS + 100 ng mL$^{-1}$ *Salmonella enterica* serovar Minnesota R595 (Millipore) and incubated at 37 °C with 5% CO2. Bacteria were added at an MOI of 100:1 and were allowed to infect for 30 min, after which media was aspirated and replaced with fresh RPMI containing 100 μg mL$^{-1}$ gentamicin (to kill extracellular bacteria) for 30 min at 37 °C with 5% $CO_2$. Media was then aspirated and replaced with fresh RPMI, and macrophages were scraped into the media.

20 μL of suspended macrophages was then transferred to a hydrophilic polycarbonate 0.2 μm Millipore Isopore GTTP filter (Merck Millipore), with a Kimwipe (Kimberly-Clark Professional) underneath to remove liquid without vacuum. Filters were attached to a glass slide with an adhesive and examined using AFM. Macrophages lysed immediately upon removal of medium, with bacterial cells remaining intact for surface scanning. For bacteria grown in LPM or MHB, cells were grown to mid-log phase ($OD_{600}$ ~ 0.5), then 20 μL of the suspension was placed on the same filter as above. The liquid was removed with a Kimwipe, then 20 μL of 10 mM MES pH 5.5 was overlaid, and liquid removed. Filters were mounted on a glass slide and examined with AFM.

A Bruker BioScope Catalyst AFM, with a Nanoscope V controller, was used to scan bacterial surfaces. For each sample, a 0.65 μm thick $Si_3N_4$ triangular cantilever was used (Scan Asyst AIR, Bruker), with a symmetric tip and spring constant of ~0.4 N m$^{-1}$. All AFM was done at 25 °C (ambient room temperature), with a scan rate of 0.5 Hz and 256 samples per line resolution. Scanning was done in PeakForce quantitative nanomechanical mapping mode. All downstream image processing was done using NanoScope software (Bruker). For scans of whole cells, scans were fit to a plane to normalize the Z-Height. For scans of bacterial surface topology, images were flattened using a second order transformation to account for subtle cell curvature, and surface topography was calculated from cross sections of these image scans.

**DiSC$_3$(5) assay.** Subcultures of WT *S.* Tm or MRSA were grown to late-exponential phase ($OD_{600}$ ~1) in MHB (MRSA) or MHB with 10 mM EDTA (*S.* Tm). Gram-negative outer membrane disruption is required for the highly lipophilic 3′,3-dipropylthiadicarbocyanine iodide (DiSC$_3$(5)) to access the cytoplasmic membrane[67]. Cells were harvested by centrifugation, washed twice in buffer (5 mM HEPES, pH 7.2, 20 mM glucose), and then resuspended in buffer to a final $OD_{600} = 0.085$ with 1 μM DiSC$_3$(5). For the experiment presented in Supplementary Fig. 4B 100 mM KCl was added to the cell suspension containing DiSC$_3$(5). After a 20 min incubation at 37 °C, 150 μL of DiSC$_3$(5) loaded cells was added to two-fold dilutions of metergoline, valinomycin, or CCCP in 96-well black clear-bottom plates (Corning) and fluorescence (excitation = 620 nm, emission = 685 nm) was read 1 min later using a Tecan M1000 Infinite Pro plate reader. The fluorescence of metergoline diluted in buffer was negligible (<200 a.u.). The fluorescence intensity was stable (<5% fluctuation) for at least 15 min when the plate was shielded from light.

**Measurement of intracellular ATP levels.** WT *S.* Tm was grown in MHB with 1 mM EDTA to early-log phase ($OD_{600} = 0.2$) and then grown in the presence of metergoline or CCCP for 30 min in clear flat-bottom 96-well plates. The $OD_{600}$ was determined immediately before ATP levels were measured using a BacTiter-Glo$^{TM}$ Microbial Cell Viability Assay (Promega), according to manufacturer instructions, in a white 96-well plate using an EnVision plate reader (PerkinElmer). Relative ATP levels were calculated by dividing relative light units (RLU) by the $OD_{600}$ (RLU/OD).

**Animal infections.** Before infection, mice were relocated at random from a housing cage to treatment or control cages. Six- to ten-week-old female C57BL/6 mice (Charles River Laboratories) were infected intraperitoneally (i.p.) with ~$10^5$ cfu *S.* Typhimurium SL1344 in 0.1 M Hepes (pH 8.0) with 0.9% NaCl. Metergoline was administered at 5 mg kg$^{-1}$ via i.p. injection, solubilized in 5% DMSO in DMEM. 5% DMSO in DMEM was given to vehicle control groups of mice. Metergoline treatments were administered every 12 h for the duration of infection, beginning at either the time of infection or 12 h post infection, as indicated per experiment. Clinical endpoint was determined using body condition scoring analyzing weight loss, reduced motility, and hunched posture. Experimental endpoint was defined as 60 h post infection for cfu comparison experiments; at this time point *S.* Tm-infected mice have undergone ~10–12% weight loss and display signs of clinical illness. At experimental endpoint, mice were euthanized, and the spleen, liver, cecum, and colon were aseptically collected into ice-cold PBS and homogenized. Bacterial load in each tissue type was enumerated from organ homogenates serially diluted in PBS and plated onto solid LB supplemented with 100 μg mL$^{-1}$ streptomycin.

**Serial passage in the presence of metergoline.** Freshly streaked colonies of MRSA or Δ*tolC S.* Tm (three colonies for each strain) were grown overnight in MHB. Each culture was diluted 500-fold into MHB and 150 μL of each sample was added to 2-fold dilutions of metergoline in a 96-well plate. After 24 h incubation at 37 °C with shaking, the well with the highest concentration of metergoline and visible growth (defined as 1/2× MIC) was diluted 500-fold into fresh MHB and the assay repeated.

**Statistical analysis.** Data were analyzed using RStudio version 1.0.143 with R version 3.2.2, GraphPad Prism 6.0 software (GraphPad Inc., San Diego, CA). Specific statistical tests and multiple test corrections are indicated in figure legends. $P$ values of <0.05 were considered significant.

**Reporting Summary**. Further information on experimental design is available in the Nature Research Reporting Summary linked to this Article.

## Data availability

The source data underlying Figs. 1a, b, 2a, b, 3a, 4 are provided as Supplementary Data 1-7. All other data are available from the corresponding authors by request.

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

## Acknowledgements

We are grateful to Susan McCusker for excellent technical assistance in high-throughput compound screens, and Maya Farha for useful discussions throughout this work. M.J.E. was supported by a post-doctoral Fellowship from the Canadian Institutes of Health Research (CIHR); C.N.T. was supported by a Canada Graduate Scholarship from the Natural Sciences and Engineering Research Council and an Ontario Graduate Scholarship. This work was supported by operating grants from CIHR to B.K.C. (388221 and 376674) and E.D.B. (FRN-143215), the Canada Research Chairs program (to B.K.C. and E.D.B.), the Ontario Research Fund (RE07-048 to B.K.C., and E.D.B.), and a donation from the Boris Family Fund for Health Research Excellence (to B.K.C. and E.D.B.). J.W.J. and J.M. were supported by a startup grant from the McMaster Faculty of Health Sciences Dean's Fund. M.M., S.P. and H.A.-P. were supported in part by NIH Grant R01AI075093, and NIAID Contract No. HHSN272200900040C.

## Author contributions

M.J.E., C.N.T., B.K.C., and E.D.B. conceived and designed the research. J.W.J. and J.M. designed metergoline analogs. M.J.E. and C.N.T. performed all experiments and analyzed data with the following exceptions: J.W.J. synthesized and analyzed metergoline analogs, S.F. performed atomic force microscopy, W.E. assisted with tissue culture. S.P., H.A-P., and M.M. constructed and supplied the *Salmonella* deletion library. J.M., B.K.C., and E.D.B. supervised research. M.J.E., C.N.T., B.K.C., and E.D.B. wrote the paper. All authors commented on the manuscript.

## Additional information

**Competing interests:** The authors declare no competing interests.

