## [Peer Review File · Nature Communications]

Reviewers' comments:

Reviewer #1 (Remarks to the Author):

The manuscript by Ellis et al. describes a parallel chemical screen aimed at identifying compounds active against intracellular bacteria. The authors surprisingly found that a drug thought to be primarily active against Gram positive bacteria is active against intracellular *Salmonella* Typhimurium. The antibiotic metergoline is active against *Salmonella* when the outer membrane is permeabilized within macrophages or when the outer membrane is perturbed in vitro. Metergoline significantly prolonged animal survival in an typhoid model of *Salmonella* Typhimurium infection.

The findings of the presented manuscript are novel and well presented. They add a new compound to the list of drugs active against *Salmonella*. The experiments were carefully designed and rigorously performed. However, the following points should be addressed:

- 1) When was the metergoline treatment started - before or after infection? If metergoline was given at the time of infection or even before: authors should show that it also works as a therapeutic, e.g. 12h or 1 day after infection. Metergoline treatment regimen should be detailed in MM and not only in the figure legend.
- 2) The authors do not comment on any effects of metergoline on mice. Toxicity in vitro was tested only on one type of cell. How does the treatment affect the whole organism, i.e. a non-infected mouse? Did the authors note any adverse effects?
- 3) The authors might want to consider adding an experiment with outbred mice and/or mice from a different vendor to exclude genotype/microbiota specific effects.
- 4) Does metergoline synergize with polymyxin B also in vivo in the mouse? Extracellular bacteria might be rendered susceptible and survival of mice further enhanced.
- 5) Generation of the analogues presents a tremendous amount of work but does not add much to the conclusion of the paper. The authors might consider moving most of the analogues in Fig. 5 to the supplement in favor of some of the many supplementary figures they are referring to that are relevant to the manuscript (e.g. S3 and S4b mentioned below, Fig. S2a and potentially others should be moved to a main figure). The analogues did not surpass the effectivity of metergoline and were not even tested in vivo (except as a control).
- 6) The supplementary picture S3 is easier to understand than the actual Fig. 4c. The 2D pictures on the right should be moved to the main figure. The control in MHB also needs to be added as a reference for smooth surface topology. In the right picture of 4c it is difficult to estimate the distance between valley and peak. Is "0" the deepest point of the valley? How was the 50% increase in average surface roughness for intracellular *Salmonella* to those grown in MHB calculated?

Minor:

- 7) Figure 4d needs quantification. Figures 4d-f need statistical analysis
- 8) Fig. S4b should be moved to the main figure, as Fig. S4 has otherwise only *S. aureus* data and S4b would fit well with the *Salmonella* data in the main figure.
- 1) Grammar lines 174-177 "deletion of genes"... "are required for normal growth"

Reviewer #2 (Remarks to the Author):

This manuscript describes a search for compounds that are specifically active against *Salmonella* residing intracellularly in macrophages. The authors describe the phenotyping of a large *Salmonella* mutant collection in LPM medium, which is assumed to mimic conditions that *Salmonella* encounter in intracellular *Salmonella*-containing vacuole. They find that some of mutants with poor growth in LPM are also impaired in macrophages. They then use a chemical screen in LPM and macrophages to identify a specifically intracellular anti-*Salmonella* activity of the

psychoactive substance metergoline. They link this intracellular activity to the low pH of the Salmonella-containing vacuole. They also demonstrate low but significant anti-Salmonella activity in a mouse infection model.

Commonly used screening conditions for finding new antimicrobials are highly artificial. Attempts to develop conditions that are closer to relevant in vivo environments are highly important. The specific activity of metergoline against intracellular Salmonella is highly interesting. On the other hand, similar screens have been already carried out in the past, the mechanistic insights are largely inconclusive, and the proposal to use LPM medium as a surrogate for in vivo conditions is not supported by the data.

1. Suitability of LPM

Parameters of LPM such as low pH, low magnesium and phosphate content, and limited nutrient availability are commonly proposed to be representative of conditions in the Salmonella-containing vacuole. However, data shown in Fig 2b demonstrate extensive discordance in compound activity in LPM vs. macrophages, and no better correlation than for a standard in vitro medium (MOPS). This suggests that LPM is not particularly predictive for anti-Salmonella activity in vivo.

2. role of low pH

The authors propose that low pH disrupting the outer membrane barrier is a key parameter for activity of metergoline in the Salmonella-containing vacuole. However, the described evidence is inconclusive. Bafilomycin prevents vacuole acidification, but this has pleiotropic effects on vacuole maturation and vesicle trafficking, and which of these many effects is actually promoting metergoline activity as well as surface topology remains unclear.

LPM is acidic (pH 5.8) but metergoline is poorly active, and if pH modulates this weak activity in LPM is not directly tested. Instead media with multiple differences in composition (LPM, MOPS, MHB) are compared and the role of pH remains unclear.

The synergism with bicarbonate (Fig 4c) is minimal, and that this minor effect is due to inner membrane perturbations is not proven.

The breakdown of membrane potential and ATP (Fig 4e,f) could be causative, or simply the indirect consequence of death. There is also no positive control for lack of membrane potential.

Although OM weakening increases metergoline potency in MHB, it remains unclear if this is the key factor also in SCVs. This is maybe unlikely, as Salmonella actually reinforce the OM by multiple PhoP-dependent modifications. Alternatively, a target that is inhibited by metergoline might be more important for Salmonella growth inside cells compared to in vitro media.

Minor points:

Why was the genetic interaction screen (Fig. 4) done in MHB and not in LPM medium?

The toxicity scale in Fig 2b is confusing: does the color for metergoline indicate that it causes 60% toxicity? How does this fit to the exclusion criterion of <25% ?

Fig. 6: The similar efficacy in systemic tissues and caecum/colon is very surprising. Is metergoline also active against extracellular Salmonella in vivo?

Line 44: "macrolides"; I guess you rather mean cephalosporines

Reviewers' comments:

Reviewer #1 (Remarks to the Author):

The manuscript by Ellis et al. describes a parallel chemical screen aimed at identifying compounds active against intracellular bacteria. The authors surprisingly found that a drug thought to be primarily active against Gram positive bacteria is active against intracellular *Salmonella Typhimurium*. The antibiotic metergoline is active against *Salmonella* when the outer membrane is permeabilized within macrophages or when the outer membrane is perturbed in vitro. Metergoline significantly prolonged animal survival in a typhoid model of *Salmonella Typhimurium* infection.

The findings of the presented manuscript are novel and well presented. They add a new compound to the list of drugs active against *Salmonella*. The experiments were carefully designed and rigorously performed. However, the following points should be addressed:

Response: We thank the reviewer for their positive comments on our manuscript.

1) When was the metergoline treatment started - before or after infection? If metergoline was given at the time of infection or even before: authors should show that it also works as a therapeutic, e.g. 12h or 1 day after infection. Metergoline treatment regimen should be detailed in MM and not only in the figure legend.

Response: Metergoline treatment was administered beginning at the time of infection. We have now added more detail regarding the treatment regimen for metergoline to both the methods as well as the main text (lines 302-312), and added additional experiments (Figure S5) confirming metergoline efficacy when treatments were administered starting at 12 hours post-infection.

2) The authors do not comment on any effects of metergoline on mice. Toxicity in vitro was tested only on one type of cell. How does the treatment affect the whole organism, i.e. a non-infected mouse? Did the authors note any adverse effects?

Response: We agree that this data should be included in the manuscript. We have added a comment in the manuscript (lines 299-301) to include our observations of weight loss in uninfected, metergoline-treated mice, which became milder as we decreased dose. These initial dosing studies helped inform the rationale behind the in vivo dose we ultimately settled on, which we have explained in the revised manuscript.

3) The authors might want to consider adding an experiment with outbred mice and/or mice from a different vendor to exclude genotype/microbiota specific effects.

Response: We thank the reviewer for this suggestion. Although genotype/microbiota effects are important to consider in animal infection experiments, we have chosen to use the well-established C57BL/6 mouse model for our *Salmonella* infections, as the course of infection has been most robustly characterized in this model system, and less-so in outbred models such as CD1. Without a rigorous characterization of microbiota and genotypes of different strains, which

is beyond the scope of our study, we feel this information would not significantly extend our conclusions.

4) Does metergoline synergize with polymyxin B also *in vivo* in the mouse? Extracellular bacteria might be rendered susceptible and survival of mice further enhanced.

Response: We agree that this is an interesting question. We performed *in vivo* experiments to test for synergy between polymyxin B and metergoline (see below data), as well as with other extracellular-active antibiotics such as colistin and ciprofloxacin. We did observe mild synergy between metergoline and polymyxin B in the liver in this preliminary experiment. However, wild type *S. Tm* strain SL1344 (for which the intraperitoneal systemic infection model was established) is generally susceptible to all therapeutically relevant antibiotics. Indeed, we found that administering polymyxin B at 1 mg/kg (several-fold lower than the equivalent human therapeutic dose) was effective on its own in enhancing survival of mice. A rigorous understanding of synergistic activity between metergoline and other antibiotic would require *in vivo* experiments with alternative drug-resistant *S. Tm* strains. While this would certainly be an interesting question to explore, it is beyond the scope of this manuscript. Given the preliminary nature of these combination data, we include them below for the purpose of review but have chosen not to include them in the final version of the manuscript.

5) Generation of the analogues presents a tremendous amount of work but does not add much to the conclusion of the paper. The authors might consider moving most of the analogues in Fig. 5 to the supplement in favor of some of the many supplementary figures they are referring to that are relevant to the manuscript (e.g. S3 and S4b mentioned below, Fig. S2a and potentially others should be moved to a main figure). The analogues did not surpass the effectivity of metergoline and were not even tested *in vivo* (except as a control).

Response: We agree with this feedback and have made the appropriate changes to the manuscript. We now include only a small figure panel (Figure 6B) in the main text, highlighting

two analogs with particularly striking *in vitro* potency, and have moved the remainder of the analog data to a supplementary figure (Figure S6).

6) The supplementary picture S3 is easier to understand than the actual Fig. 4c. The 2D pictures on the right should be moved to the main figure. The control in MHB also needs to be added as a reference for smooth surface topology. In the right picture of 4c it is difficult to estimate the distance between valley and peak. Is “0” the deepest point of the valley? How was the 50% increase in average surface roughness for intracellular Salmonella to those grown in MHB calculated?

Response: As requested, we moved the supplementary AFM figure into the main text (Figure 4), which now includes the appropriate MHB controls and 2D images. We also added more detail to the section of the manuscript describing the AFM data to clarify the valley-peak measurements.

Minor:

7) Figure 4d needs quantification. Figures 4d-f need statistical analysis

Response: We included photographs of these cultures rather than a graph with quantitation of culture turbidity as this is a commonly used way of showing that a compound is bacteriolytic. For the reviewer’s reference, we have included (below) a representative kinetic lysis experiment with metergoline against EDTA treated cells. We have added information about statistical analysis to the relevant figures.

8) Fig. S4b should be moved to the main figure, as Fig. S4 has otherwise only *S. aureus* data and S4b would fit well with the *Salmonella* data in the main figure.

Response: As recommended by the reviewer, we have moved Figure S4B to Figure 5C.

1) Grammar lines 174-177 “deletion of genes”... “are required for normal growth”

Response: We have reworded this sentence for clarity.

Reviewer #2 (Remarks to the Author):

This manuscript describes a search for compounds that are specifically active against *Salmonella* residing intracellularly in macrophages. The authors describe the phenotyping of a large *Salmonella* mutant collection in LPM medium, which is assumed to mimic conditions that *Salmonella* encounter in intracellular *Salmonella*-containing vacuole. They find that some of mutants with poor growth in LPM are also impaired in macrophages. They then use a chemical screen in LPM and macrophages to identify a specifically intracellular anti-*Salmonella* activity of the psychoactive substance metergoline. They link this intracellular activity to the low pH of the *Salmonella*-containing vacuole. They also demonstrate low but significant anti-*Salmonella* activity in a mouse infection model.

Commonly used screening conditions for finding new antimicrobials are highly artificial. Attempts to develop conditions that are closer to relevant *in vivo* environments are highly important. The specific activity of metergoline against intracellular *Salmonella* is highly interesting. On the other hand, similar screens have been already carried out in the past, the mechanistic insights are largely inconclusive, and the proposal to use LPM medium as a surrogate for *in vivo* conditions is not supported by the data.

1. Suitability of LPM

Parameters of LPM such as low pH, low magnesium and phosphate content, and limited nutrient availability are commonly proposed to be representative of conditions in the *Salmonella*-containing vacuole. However, data shown in Fig 2b demonstrate extensive discordance in compound activity in LPM vs. macrophages, and no better correlation than for a standard *in vitro* medium (MOPS). This suggests that LPM is not particularly predictive for anti-*Salmonella* activity *in vivo*.

Response: We agree with the reviewer that LPM is an imperfect mimic of the SCV and have carefully revised our manuscript to reflect this. The points we now emphasize are the expansion of target space accessible by the use of LPM relative to conventional nutrient-rich growth media (i.e., MHB), and the need to conduct preliminary screening efforts outside of cell culture due to limitations in throughput. Although beyond the scope of the current manuscript, we have observed in other screens with more diverse chemical libraries that compounds active in LPM and not MHB are more likely to have activity against intracellular bacteria.

2. role of low pH

The authors propose that low pH disrupting the outer membrane barrier is a key parameter for activity of metergoline in the *Salmonella*-containing vacuole. However, the described evidence is inconclusive. Bafilomycin prevents vacuole acidification, but this has pleiotropic effects on vacuole maturation and vesicle trafficking, and which of these many effects is actually promoting metergoline activity as well as surface topology remains unclear. LPM is acidic (pH 5.8) but metergoline is poorly active, and if pH modulates this weak activity in LPM is not directly tested. Instead media with multiple differences in composition (LPM, MOPS, MHB) are compared and the role of pH remains unclear.

Response: We are grateful to the reviewer for these comments as it prompted us to perform additional experiments that revealed that low Mg^{2+} , and not low pH, is responsible for OM permeabilization in LPM. We have now made significant changes to the text and added some key experiments to clarify the component of LPM/macrophages that permits metergoline activity. In acknowledging the pleiotropic effects of bafilomycin pre-treatment on endosomal trafficking within macrophages, we elected to remove these data from the manuscript, as we believe that the intramacrophage and AFM experiments were misleading. We have added experiments demonstrating instead a Mg^{2+} -dependent shift in the MIC of metergoline against *S. Tm* grown in LPM (Figure 3E, Figure S2C) accompanied by increased NPN uptake (Figure 3F). Interestingly, we saw a similar response with 3 other antibiotics antagonized by the OM (vancomycin, rifampicin, mupirocin). We believe that these data provide more conclusive evidence of the role of low Mg^{2+} , and not low pH, in both permitting the conditional activity of metergoline in LPM and/or macrophages, and in disrupting the OM integrity of intracellular *S. Tm*.

The synergism with bicarbonate (Fig 4c) is minimal, and that this minor effect is due to inner membrane perturbations is not proven.

Response: The impact of bicarbonate on antibiotic activity was first reported last year (Ersoy et al, 2017, EBioMedicine) and we published work this year demonstrating that bicarbonate alters the proton motive force by collapsing ΔpH (Farha et al, 2018, ACS Infect Dis). We agree that this manuscript does not prove the modest effect of bicarbonate on metergoline is due to inner membrane perturbation, but when combined with published work and our other data, we believe that this is the most likely explanation. We have been careful in the manuscript to not overstate these conclusions.

The breakdown of membrane potential and ATP (Fig 4e,f) could be causative, or simply the indirect consequence of death. There is also no positive control for lack of membrane potential.

Response: We agree that effect of metergoline on cellular ATP might be an indirect consequence of death, but we favour the alternative explanation that ATP levels are reduced as a consequence of altered PMF. ATP levels were measured after a short (30 min) incubation with metergoline and were normalized to optical density of the cultures. We have also observed that metergoline inhibits swimming motility without affecting flagella biosynthesis, which is fully consistent with a decrease in available energy (we elected not to include these data in the manuscript in the interest of maintaining clarity). To address the issue of controls brought up by the reviewer, we have included DiSC3(5) assays with valinomycin (Figure S4B) and CCCP (Figure S4C), which are controls for decreased and increased membrane potential, respectively.

Although OM weakening increases metergoline potency in MHB, it remains unclear if this is the key factor also in SCVs. This is maybe unlikely, as *Salmonella* actually reinforce the OM by multiple PhoP-dependent modifications. Alternatively, a target that is inhibited by metergoline might be more important for *Salmonella* growth inside cells compared to in vitro media.

Response: The reviewer raises two very good points that we have addressed in the revised manuscript. We emphasized that the activity of metergoline against *S. aureus* (a Gram-positive bacteria) suggests that metergoline has a conserved target, and not a *Salmonella*-specific protein required only in the SCV. We have made substantial changes to the discussion concerning the presumed role of PhoP in OM stress. To our knowledge, there is no work directly measuring the impact of phosphoethanolamine or 4-aminoarabinose modifications to lipid A on OM integrity in the SCV. Work from the Nikaido and Miller labs (Murata et al, 2007, J Bacteriol) found that constitutively active PhoP increased OM integrity but these experiments were performed in LB and in a $\Delta tolC$ genetic background, making a direct comparison to our work difficult. PhoP-dependent lipid A modifications alter OM charge and confer resistance to cationic antimicrobial peptides, but we are not sure how these modifications alter OM permeability to small molecules. Indeed, expression of the *mcr-1* encoded phosphoethanolamine transferase is proposed to increase OM permeability leading to collateral sensitivity to tigecycline and other antibiotics (Yang et al, 2017, Nature Communications). We hope that our work will prompt further research into this interesting paradox in *Salmonella* physiology.

Minor points:

Why was the genetic interaction screen (Fig. 4) done in MHB and not in LPM medium?

Response: We have added a short explanation for this in the relevant section of the results (lines 166-168). Briefly, we conducted the chemical-genetic screen to understand why metergoline was inactive in MHB, which we thought would highlight intrinsic resistance mechanisms to this compound in rich media.

The toxicity scale in Fig 2b is confusing: does the color for metergoline indicate that it causes 60% toxicity? How does this fit to the exclusion criterion of <25% ?

Response: We thank the reviewer for catching this mistake. We have now corrected the scale in the heatmap and made modifications to the colouring for ease of interpretation.

Fig. 6: The similar efficacy in systemic tissues and caecum/colon is very surprising. Is metergoline also active against extracellular *Salmonella* in vivo?

Response: We agree that this is a surprising result. At present we can only speculate about the source of this result, but we hypothesize that the process of reseeded in the gastrointestinal tract from the liver (via the hepatic biliary system) following intraperitoneal injection exerts unique stresses on *S. Tm* cells *in vivo* to further increase the conditional sensitivity to metergoline. Indeed, several groups have investigated the conditions within the gallbladder and bile duct during *S. Tm* colonization and persistence in these niches. For clarity, we added a brief interpretation of the cecum/colon treatment data into our Discussion.

Line 44: “macrolides”; I guess you rather mean cephalosporines

Response: We thank the reviewer for catching this and have made the appropriate correction in the manuscript.

REVIEWERS' COMMENTS:

Reviewer #1 (Remarks to the Author):

The authors have substantially improved the manuscript. Significant passages of the paper have been rewritten and clarity of the manuscript has benefitted from it. The authors have also sufficiently addressed my concerns regarding additional experiments to support their claims. In particular, metergoline-treated uninfected mice and the therapeutic model are important additions to the manuscript. The effect of metergoline in the therapeutic model is less striking than when given at the time of infection, but still significant. The authors now describe in detail the effects of metergoline on uninfected mice.

I only have one question remaining:

Supplementary Figure 2: Phosphate addition should suppress metergoline activity according to the text. However, the line with phosphate addition seems to overlap with the LPM 5.8 line, indicating that phosphate addition has no effect on metergoline activity. It is difficult to see, as the light grey line is in the background. Can the authors check this graph and/or move the light grey line into the foreground?

REVIEWERS' COMMENTS:

Reviewer #1 (Remarks to the Author):

The authors have substantially improved the manuscript. Significant passages of the paper have been rewritten and clarity of the manuscript has benefitted from it. The authors have also sufficiently addressed my concerns regarding additional experiments to support their claims. In particular, metergoline-treated uninfected mice and the therapeutic model are important additions to the manuscript. The effect of metergoline in the therapeutic model is less striking than when given at the time of infection, but still significant. The authors now describe in detail the effects of metergoline on uninfected mice.

Response: We thank the reviewer for their positive comments on the revised manuscript.

I only have one question remaining:

Supplementary Figure 2: Phosphate addition should suppress metergoline activity according to the text. However, the line with phosphate addition seems to overlap with the LPM 5.8 line, indicating that phosphate addition has no effect on metergoline activity. It is difficult to see, as the light grey line is in the background. Can the authors check this graph and/or move the light grey line into the foreground?

Response: We greatly appreciate the reviewer catching this mistake. This was an error in the text, as we meant to write that Mg^{2+} addition suppresses metergoline (corresponding to the dark grey line as is currently shown in Supplementary Fig. 2).

P11 L241-242 has now been corrected to "...; for metergoline, only Mg^{2+} supplementation suppressed activity (Supplementary Fig. 2C)